# Differentially Expressed miRNAs in Age-Related Neurodegenerative Diseases: A Meta-Analysis

**DOI:** 10.3390/genes13061034

**Published:** 2022-06-09

**Authors:** Ocana Noronha, Lucia Mesarosovo, Jasper J. Anink, Anand Iyer, Eleonora Aronica, James D. Mills

**Affiliations:** 1Department of (Neuro)Pathology, Amsterdam UMC, University of Amsterdam, Amsterdam Neuroscience, 1105 AZ Amsterdam, The Netherlands; ocana.noronha@riken.jp (O.N.); l.mesarosova@amsterdamumc.nl (L.M.); j.j.anink@amsterdamumc.nl (J.J.A.); e.aronica@amsterdamumc.nl (E.A.); 2Neurodegenerative Disorders Collaborative Laboratory, RIKEN Center for Brain Science, Saitama 351-0106, Japan; 3Department of Internal Medicine, Erasmus Medicine Center, 3015 GD Rotterdam, The Netherlands; a.iyer@erasmusmc.nl; 4Department of Clinical and Experimental Epilepsy, University College London, London WC1E 6BT, UK; 5Chalfont Centre for Epilepsy, Chalfont St Peter, Gerrards Cross SL9 0RJ, UK

**Keywords:** neurodegenerative diseases, miRNAs, synucleinopathies, amyloidopathies, tauopathies

## Abstract

To date, no neurodegenerative diseases (NDDs) have cures, and the underlying mechanism of their pathogenesis is undetermined. As miRNAs extensively regulate all biological processes and are crucial regulators of healthy brain function, miRNAs differentially expressed in NDDs may provide insight into the factors that contribute to the emergence of protein inclusions and the propagation of deleterious cellular environments. A meta-analysis of miRNAs dysregulated in Alzheimer’s disease, Parkinson’s disease, multiple system atrophy, progressive supranuclear palsy, corticobasal degeneration, dementia with Lewy bodies and frontotemporal lobar degeneration (TDP43 variant) was performed to determine if diseases within a proteinopathy have distinct or shared mechanisms of action leading to neuronal death, and if proteinopathies can be classified on the basis of their miRNA profiles. Our results identified both miRNAs distinct to the anatomy, disease type and pathology, and miRNAs consistently dysregulated within single proteinopathies and across neurodegeneration in general. Our results also highlight the necessity to minimize the variability between studies. These findings showcase the need for more transcriptomic research on infrequently occurring NDDs, and the need for the standardization of research techniques and platforms utilized across labs and diseases.

## 1. Introduction

Neurodegenerative diseases (NDDs) are characterized by the gradual deterioration of the function and structure of specific cell populations within the brain, which ultimately leads to neuronal and/or glial cell death. As of 2016, neurological disorders accounted for the maximum disability-adjusted life years globally, with the most prevalent NDDs, Alzheimer’s disease (AD) and Parkinson’s disease (PD), demonstrating increased incidences of 117% and 145%, respectively, from 1990 to 2016 [1] The aggregation of misfolded proteins is a pathogenic feature observed in multiple NDDs, collectively termed proteinopathies. Accordingly, NDDs can be classified by their underlying pathological protein aggregates. For instance, tauopathies arise due to hyperphosphorylation and the misfolding of microtubule-associated protein Tau (MAPT) isoforms, which leads to its pathological accumulation in neurons or glial cells [2]. AD, corticobasal degeneration (CBD), progressive supranuclear palsy (PSP) and dementia with Lewy bodies (DLB) are examples of diseases with tau-positive neurofibrillary pathology [2,3]. α-Synucleinopathies are NDDs that are characterized by abnormal α-Synuclein protein aggregates in neurons or glial cells. PD, multiple system atrophy (MSA) and DLB are the most prevalent synucleinopathies [4]. Amyloidopathy refers to a pathological condition in which proteins misfold and demonstrate stable β-sheet secondary structures, which prevent their proteolysis [5]. Amyloid β protein accumulation in the brain is widespread in AD, and its deposition has also been observed in patients with PD and DLB [6,7,8]. TDP-43 proteinopathies are characterized by hyperphosphorylated TDP-43 intranuclear and cytoplasmic aggregates in neurons and glial cells. AD and frontotemporal lobar degeneration TDP-43 (FTLD TDP-43) are NDDs that present TDP-43 pathology [9,10]. While proteinopathies share a common pathogenesis, the vulnerable cells targeted, and the route of the prion-like progression within the brain, vary from one disease to another [7,10,11,12,13,14,15,16,17,18,19,20].

Examining transcript expression patterns and determining cellular states provide insight into the underlying molecular pathology of NDDs. As a result, multiple research groups have conducted and compared transcriptome analyses between NDD patients and healthy controls [21,22,23,24,25,26]. These analyses have yielded results that are indicative of the importance of not only coding but also noncoding RNAs (ncRNAs), including microRNAs (miRNAs), in the regulation of all biological processes. miRNAs are small ncRNAs with an average length of 22 nucleotides. They interact with the 3′ UTR and, to a lesser extent, with the 5′ UTR of target mRNA, gene promoter sequences and coding sequences to degrade mRNA, regulate transcription and repress or activate translation under specific conditions [27,28,29]. The activity of miRNA-associated protein complexes allow individual miRNAs to potentially target anywhere from tens to hundreds of different messenger RNAs (mRNAs) [30]. Correspondingly, in humans, at the translational level, miRNAs regulate the expression of approximately 60% of protein-coding genes [31], and interact, to an even greater extent, with the noncoding regions of the genome, as is inferred through target prediction tools. Due to these regulatory functions, miRNAs are critical in maintaining the homeostatic functionality of the brain. Studies focusing on miRNA profiling showcase a widespread dysregulaton of miRNAs in multiple NDDs (refer to Table 1), which is indicative of their versatile regulatory role. Since a shift in the physiological expression of miRNAs exhibits established associations with progressing pathogenesis in multiple NDDs [26,32], a wide-scale imbalance in miRNA regulation could facilitate the development of neuropathologies.

In order to gain deeper insights into the pathogenic mechanisms of proteinopathies and across neurodegeneration, we performed a systematic literature review and conducted a meta-analysis using publicly available miRNA expression datasets from seven NDDs: AD, PD, MSA, DLB, CBD, PSP and FTLD-TDP43. We aimed to assess if miRNAs were consistently differentially expressed within a singular disease, and to identify whether proteinopathies could be categorized on the basis of miRNA expression signatures. Additionally, we hypothesized that the common pathogenesis underlying proteinopathies may generate a pan-neurodegenerative miRNA expression signature (i.e., consistently differentially expressed miRNAs across NDDs). Thereafter, we sought to understand if the identified miRNAs target a convergent or divergent set of genes, which, in turn, modulate specific pathways within a disease or proteinopathy, or modulate common biological pathways across neurodegeneration. To date, multiple laboratories have focused their efforts on performing meta-analyses on the gene and miRNA expression datasets of single NDDs, such as AD [33,34,35,36,37] and PD [38,39,40,41], with some focus on the gene expression in two or more NDDs [42,43,44,45]. However, to the best of our knowledge, this is the first initiative undertaken using solely human miRNA expression data to identify consistently differentially expressed miRNAs and enriched pathways that are specific to a single proteinopathy, and to neurodegeneration as a whole.

## 2. Materials and Methods

### 2.1. Collection of Data

The collection of articles to be included in this meta-analysis was based on a systematic literature search using the PubMed database. The diseases encompassed were shortlisted on the basis of common or intertwining pathologies. Accordingly, articles containing data on the miRNA expression profiles in AD, PD, MSA, DLB, PSP, CBD and FTLD-TDP43 were screened. The specific keywords used for selecting studies were the “Disease name”, “microRNA” and the profiling technique: “RNA sequencing”, “microarray”, “transcriptome sequencing”. The studies were then screened to only include human miRNA expression profiles obtained from brain tissue or biofluids. As transcriptomic analysis can not only distinguish cells on the basis of their current states and future fates, but can also categorize them as per functionality and tissue type [46,47], studies using peripheral blood cells were excluded within this meta-analysis to avoid RNA expression reflecting cellular functions from non-nervous tissue. The last search for inclusion was performed in January 2021. While the majority of the studies included patients with sporadically arising neurodegenerative diseases, the expression profiles of patients with genetic mutations leading to diseases, and specifically to PD, AD and FTLD, were also included. A minimum of 50 miRNAs had to be profiled for a study to be included. Studies validating previously implicated miRNAs via RT-qPCR were excluded. The list of the studies included in the meta-analysis is provided in a categorically descriptive manner in Table 1.

### 2.2. Database of Differentially Expressed miRNAs in NDDs

A database of the differentially expressed miRNAs in each disease was created. To ascertain the basis of commonalities observed across the different selected neurodegenerative diseases, the miRNAs were arranged in a table with respect to the disease type and pathology. A significance level of 0.05 for the differentially expressed miRNAs was accepted. The required lists of differentially expressed miRNAs and their significance levels for each study were obtained online. Authors were contacted directly if the necessary data were not readily available.

### 2.3. Robust Rank Aggregation

Next, to differentiate between experimental noise and robust evidence between and across all the selected neurodegenerative diseases, the significantly differentially expressed miRNAs were tabulated with respect to the disease type, anatomical regions and pathology, and the robust-rank-aggregation (RRA) method was implemented. Each miRNA list prepared was ranked as per its *p*-value. The RRA algorithm was applied to each list. The RRA method monitors the entire list of miRNA provided, and assigns the miRNA ranked consistently better a higher significance score under the null hypothesis [48]. Each miRNA is assigned a significance score that is indicative of its ranking. miRNAs with a permutation *p*-value < 0.05 (100,000 permutations) were considered statistically significant. The statistical software R (version 3.6.1) was used for all the analyses conducted within this study [49].

### 2.4. Gene Set Enrichment Analysis

Post RRA, a gene set enrichment analysis was performed with the miRNAs deemed consistently significant. The validated and probable interactions of the 3′-UTR of the significant miRNAs having a target prediction score of a minimum of 0.95 via the TarPmiR algorithm were identified using the miRWalk 3.0 platform [50]. Each list of miRNA-interacting genes obtained for each RRA analysis were individually enriched for pathway associations against Gene Ontology for Biological Processes (GO.BP), Cellular Components (GO.CC) and Molecular Functions (GO.MF), against all immunogenic signature gene sets, cell-type signature gene sets and KEGG, and Reactome gene sets obtained from the Molecular Signatures Database (MsigDB) [51,52]. The enrichment was performed using the run enrichment function with true composite, thus treating each miRNA-gene association as a separate enriched entity [53].

**Table 1 genes-13-01034-t001:** List of Neurodegenerative diseases included in the meta-analysis categorized by disease.

Sr. No.	Study	Disease	Platform	Profiled	Criteria	Anatomy
1	Schulze et al., 2018 [54]	PD	Illumina deep sequencing (TruSeq SBS v3 Kit)/GRCh38 (annotation based on ElDorado 6–2015)	1917 (92 DE)	p-adj. < 0.05; log2FC ≥ 0.6	Brain: cingulate gyrus
2	Hoss et al., 2016 [55]	PD	Illumina HiSeq 2000	1223 (124 DE)	FDR q < 0.05	Brain: prefrontal cortex (Brodmann Area 9)
3	Wake et al., 2016 [56]	PD	Illumina HiSeq 2000	2584 (3 DE)		Brain: prefrontal cortex (Brodmann Area 9)
4	Ding et al., 2016 [57]	PD	Illumina solexa sequencing	1123 (15 DE)	*p* < 0.001; FC ≥ 1.5; 300 ≥ copies;	Serum
5	Dong et al., 2016 [58]	PD	Illumina solexa sequencing	721 (12 DE)	FC ≥ 3; 100 ≥ copies	Serum
6	Botta-Orfila et al., 2014 [59]	PD	TaqMan Array Human MicroRNA A Cards v2.0	377 (2 DE)	*p* ≤ 0.05	Serum
7	Annese et al., 2018 [60]	PD	Illumina MiSeq platform	2589 (40 DE)	p-adj. ≤ 0.05; log2 fold change	Brain: hippocampal CA1 region
8	Burgos et al., 2014 [61]	PD	Illumina HiSeq2000	2228 (CSF: 17 DE; serum: 5 DE)	p-adj. < 0.05; 0.7 < log2 FC	CSF and serum
9	Vallelunga et al., 2014 [62]	PD	TaqMan Human MicroRNA Array v3.0 A and B (Applied Biosystems | Life Technologies™)	754 (5 DE)	*p* < 0.05	Serum
10	Gui et al., 2015 [63]	PD	TaqMan Low-Density Array Human miRNA Panels (Applied Biosystems)	746 (27 DE)	p-adj. < 0.05; FC > 2	CSF
11	Kume et al., 2018 [64]	MSA	3D-Gene miRNA oligo chips (version 17; Toray Industries, Inc.)	679 (67 DE)	*p* < 0.05	Serum
12	Lee et al., 2014 [65]	MSA-Cerebellar	Human miRNA Microarray 8 × 15 K kit (Agilent Tech)	866 (31 DE)	*p*-value < 0.05; FC > 1.5	Brain: cerebellar cortex
13	Kim et al., 2019 [66]	MSA-Parkinson	TaqMan microRNA Reverse Transcription Kit (Applied Biosystems)	800 (31 DE)	*p*-value ≤ 0.001; FDR ≤ 0.02	Brain: striatum
14	Wakabayashi et al., 2016 [67]	MSA-C and MSA-P	Human miRNA oligo chip (Toray Industries)	1734 (Pons: 38 DE; Cerebellum: 23 DE)		Brain: pons and cerebellum
15	Ubhi et al., 2014 [68]	MSA	OneArray^®^ Human microRNA Microarray v3	1087 (14 DE)	*p* < 0.05; FC > 2	Brain: frontal cortex
16	Uwatoko et al., 2019 [69]	MSA-C and MSA-P	3D-Gene^®^ Human miRNA oligo chip (Ver. 17.0)-TORAY Industries	1720 (79 DE)	*p* < 0.05	Plasma
17	Vallelunga et al., 2014 [62]	MSA-C and MSA-P	TaqMan Human MicroRNA Array v3.0 A and B (Applied Biosystems | Life Technologies™)	754 (5 DE)	*p* < 0.05	Serum
18	Ubhi et al., 2014 [68]	CBD	OneArray^®^ Human microRNA Microarray v3	1087 (12 DE)	*p* < 0.05; FC > 2	Brain: frontal cortex
19	Tatura et al., 2016 [70]	PSP	TaqMan Array MicroRNA A cards (Thermo Fisher Scientific)	372 (4 DE)	*p* < 0.05; log FC ≥ 1	Brain: inferior frontal gyri
20	Ubhi et al., 2014 [68]	PSP	OneArray^®^ Human microRNA Microarray v3	1087 (12 DE)	*p* < 0.05; FC > 2	Brain: frontal cortex
21	Hébert et al., 2013 [71]	DLB	Illumina GAIIx	1921 (21 DE)	*p* < 0.05	Brain: superior and middle temporal gyri (Brodmann Areas 21/22)
22	Hébert et al., 2013 [71]	FTLD	Illumina GAIIx	1921 (21 DE)	*p* < 0.05	Brain: superior and middle temporal gyri (Brodmann Areas 21/22)
23	Chen-Plotkin et al., 2012 [72]	FTLD	miRCURY LNA array version 11.0 (Exiqon)	836 (11 DE)	*p* < 0.05	Brain: frontal cortex
24	Hébert et al., 2013 [71]	AD	Illumina GAIIx	1921 (17 DE)	*p* < 0.05	Brain: superior and middle temporal gyri (Brodmann Areas 21/22)
25	Kumar et al., 2017 [73]	AD and MCI	Affymetrix GeneChip miRNA array v. 4.0	2584 (4 DE in AD; 50 DE in MCI)	*p* < 0.05; log FC ≥ 2	Serum
26	Lugli et al., 2015 [74]	AD	Illumina HiSeq2500	2589 (20 DE)	*p* < 0.05	Plasma
27	Wu et al., 2017 [75]	AD	Illumina HiSeq2500 Sequencer	2042 (72 DE)	*p* < 0.05; log FC ≥ 2	Serum
28	Hara et al., 2017 [76]	AD	Illumina Genome Analyzer IIx (GAIIx)	2584 (serum: 20 DE; Temporal cortex: 213 DE)	p-adj. ≤ 0.05	Serum and Brain: temporal cortex
29	Annese et al., 2018 [60]	AD	Illumina MiSeq platform	2589 (40 DE)	p-adj. ≤ 0.05; log2 fold change	Brain: (1) hippocampal CA1 region; (2) middle temporal gyrus (Brodmann Area 21); and (3) the middle frontal gyrus (Brodmann Area 46)
30	Nunez-Iglesias et al., 2010 [77]	AD	2042 (30 DE)	470 (48 DE)	FDR < 0.05	Brain: parietal lobe cortex
31	Patrick et al., 2017 [78]	AD	NanoString nCounter assay	2042 (30 DE)	p-adj. < 0.05	Brain: dorsolateral prefrontal cortex (Brodmann Areas 9 and 46)
32	Wang et al., 2011 [79]	AD	miRCURY™ LNA array version 11.0 (Exiqon, Denmark)	904 (113 DE)	*p* < 0.05	Brain: superior and middle temporal gyri (Brodmann Areas 21 and 22)
33	Hébert et al., 2008 [80]	AD	mirVana miRNA Bioarrays V2 (Ambion Inc.)	328 (16 DE)	*p* < 0.05	Brain: cerebral cortex
34	Van Harten et al., 2015 [81]	AD	qRT-PCR (Taqman Array MicroRNA cards A and B, v3.0)	754 (24 DE)	*p* < 0.05	CSF
35	Dong et al., 2015 [82]	AD	Illumina’s Solexa Sequencer	721 (4 DE)	*p* < 0.05	Serum
36	Lv et al., 2018 [83]	AD	MiRCURY™ Array (v.18.0) (Exiqon)	1223 (28 DE)	p-adj. < 0.05; FC > 2	CSF
37	Lau et al., 2013 [84]	AD	1. nCounter Human miRNA Expression Assay Kit version 1 (Nanostring Technologie); and 2. Illumina HiSeq 2000 system	641 (expression array) (Hippocampus: 35 DE); Prefrontal cortex: 41DE) and 2038 (Deep seq) (85 DE)	p-adj. < 0.05 (nCounter array); p-adj. < 0.001 (Deep seq)	Brain: (1) hippocampus; (2) prefrontal cortex
38	Ubhi et al., 2014 [68]	AD	OneArray^®^ Human microRNA Microarray v3	1087 (13DE)	*p* < 0.05; FC > 2	Brain: frontal cortex
39	Burgos et al., 2014 [61]	AD	Illumina HiSeq2000	2228 (CSF: 41 DE; serum: 20 DE)	p-adj. < 0.05; log2FC > 0.7; normalized mean > 5 mapped reads for each group	CSF and serum
40	Gui et al., 2015 [63]	AD	TaqMan Low-Density Array Human miRNA Panels (Applied Biosystems)	746 (7 DE)	p-adj. < 0.05; FC > 2	CSF

## 3. Results

Data from a total of 30 research articles were included in this study. While some articles solely focused on the differential expression of miRNAs within a single disease, many studies included in this meta-analysis presented data on multiple neurodegenerative diseases (Table 1). A database of the differentially expressed (DE) miRNAs in each NDD (namely, PD, MSA, CBD, PSP, DLB, FTLD-TDP43 and AD) passing the selected significance level of 0.05 was prepared. For each NDD, the average number of DE miRNAs per study was calculated. PD and AD presented the highest number (on average) of upregulated DE miRNAs (17 ± 21). However, MSA profiles revealed the highest number (on average) of downregulated miRNAs (25 ± 25). The general trend showed a greater number of downregulated miRNAs in neurodegeneration, with an overall average of 17 downregulated to 14 upregulated miRNAs per profile (Figure 1). A greater variability in the number of DE miRNAs in an NDD correlated with an increasing number of studies contained within the dataset for the specific NDD.

### 3.1. Heatmap Analysis

From the 30 research papers, a total of 1105 unique DE miRNAs (*p* < 0.05) were identified. To determine if a common set of miRNAs were dysregulated across a disease, an unsupervised hierarchical clustering analysis was performed. A total of 45 expression profiles were derived from the included studies for upregulated miRNAs, and 47 expression profiles for downregulated miRNAs. These expression profiles were obtained on the basis of the type of NDD and the biomaterial (cortical brain matter, noncortical brain matter or biofluid) used for identification of the dysregulated miRNAs. Of the 92 total expression datasets analyzed (45 upregulated, 47 downregulated), 53 (27 upregulated, 26 downregulated) were obtained from the brain, and 39 (18 upregulated, 21 downregulated) from biofluid specimens. The datasets included in the brain meta-analyses were obtained from different brain regions, including the cerebral cortex, cingulate gyrus, hippocampus, cerebellar cortex and striatum, while the expression datasets of only the CSF and serum samples were considered for the biofluid analyses. Collectively, 530 unique upregulated miRNAs and 575 unique downregulated miRNAs were detected across all the NDDs.

The heatmaps for the downregulated miRNAs revealed scarce numbers of miRNAs being repeatedly DE (hsa-miR-132-5p, hsa-miR-132-3p, hsa-miR-212-3p, hsa-miR-129-5p, hsa-miR-127-3p and hsa-miR-433 were downregulated in five profiles or more; hsa-miR-19b-3p, hsa-miR-212-5p, hsa-miR-26b, hsa-miR-20b, hsa-miR-127-5p, hsa-miR-485-5p, hsa-miR-129-2-3p, hsa-miR-370, hsa-miR-409-5p, hsa-miR-409-3p, hsa-miR-136-3p, hsa-miR-139-5p, hsa-miR-375 and hsa-miR-128-3p were downregulated in four profiles), whereas the heatmap for the upregulated miRNAs revealed a more distributed expression across the profiles, with only hsa-miR-24 and hsa-miR-324-3p being expressed in a maximum of four profiles. The cluster analyses largely demonstrated no consistency in the miRNAs detected across the different expression profiles of the same disease. Grouping was neither observed by disease type nor proteinopathy, with the maximum miRNAs clustering on the basis of the anatomical region from which they were extracted (Figure 2A,B). Overall, the included studies produced varied expression profiles, with no clear clustering as per the specified parameters.

### 3.2. Differential Expression of miRNAs within Proteinopathies

To identify the similarities and differences between the miRNA expression profiles of diseases, the extent of the overlap between DE miRNAs was assessed, irrespective of the tissue/biofluid from which they were identified. Firstly, diseases with a common underlying protein pathology were overlapped to ascertain if specific sets of miRNAs were dysregulated across a specific proteinopathy.

Amyloidopathies: As mentioned above, amyloidopathy is observed in PD, DLB and AD (Figure 3A,B). The number of miRNAs downregulated was greater than that of the upregulated miRNAs across the diseases. AD and PD had a greater number of DE miRNAs compared to DLB. Consequently, AD and PD displayed a 13% overlap, sharing 55 commonly downregulated miRNAs, and a 7% overlap with 30 upregulated miRNAs. Contrastingly, only 1% of the downregulated miRNAs were commonly expressed between PD and DLB, while overlaps of 1.6% and 0.3% were observed between AD and DLB in down- and upregulated miRNAs, respectively. This minimal overlap between DLB and other amyloidopathies can be attributed to a single study included in the DLB analysis.

Synucleinopathies: The synucleinopathies MSA, PD and DLB displayed a greater number of differentially downregulated miRNAs compared to upregulated miRNAs. MSA and PD showed overlaps of 6.9% and 4% between studies for downregulated and upregulated miRNAs, respectively (Figure 3C,D). Though no miRNAs were commonly upregulated across all the synucleinopathies (Figure 3C), 22 miRNAs were commonly downregulated between MSA and PD, but only one miRNA, hsa-miR-433, was downregulated in all the synucleinopathies (Figure 3D). The lone study in the DLB dataset may have contributed to the sole miRNA being commonly DE within this proteinopathy.

Tauopathies: No miRNAs were commonly DE between all the tauopathies included in the analyses: CBD, PSP, DLB and AD (Figure 3E,F). DLB only shared miRNAs with AD. As the DLB patients included in the study only showed Braak stages I and II, the absence of a common miRNA signature in the tauopathies may have resulted from the deficiency of tau pathology in the analyzed tissue. Although CBD and PSP target the same neuronal populations, no overlap was discovered between the two diseases for upregulated miRNAs (Figure 3E), whereas one miRNA was commonly downregulated (Figure 3F).

TDP-43 proteinopathies: FTLD-TDP43 was the primary TDP-43 proteinopathy included in the meta-analysis. Four common downregulated and three common upregulated miRNAs were detected between AD and FTLD-TDP43 (Figure 3G,H).

Overall, the Venn diagrams demonstrated a maximum overlap between the dissimilar Alzheimer’s and Parkinson’s diseases, which may be attributed to a greater number of profiles obtained from these diseases.

### 3.3. Differential Expression of miRNAs across Proteinopathies

Next, we aimed to identify if unique sets of miRNAs were DE within a specific proteinopathy, or were commonly expressed in neurodegeneration, irrespective of the tissue/biofluid from which they were identified. In total, 317, 431, 334 and 313 unique miRNAs were downregulated, and 298, 357, 223 and 402 miRNAs were upregulated in the tauopathies, amyloidopathies, synucleinopathies and TDP43-proteinopathies, respectively (Figure 4A,B). Across all the pathologies, 87 miRNAs were downregulated, while only 30 were upregulated.

DE miRNAs in cortical and noncortical tissue and in biofluids were overlapped to assess if miRNAs were dysregulated across the spectrum of diseases in a tissue-specific manner. A greater overlap was detected in downregulated miRNAs. Only ~3% of the downregulated miRNAs were common to all biomaterial (Figure 4D). Their differential expression was seen across multiple diseases and was not specific to a type of proteinopathy. The differentially upregulated miRNAs displayed a substantially reduced degree of overlap compared to the downregulated miRNAs (Figure 4C,D). Overall, no clear grouping as per disease type or anatomical region was observed.

*Robust Rank Aggregation*: The robust-rank-aggregation method was utilized to identify the miRNAs that were consistently differentially expressed across the studies. With a criterion of an adjusted permutation *p*-value of <0.05, 100,000 permutations were performed for the analyses for up- and downregulated miRNAs with respect to the disease type, pathological protein inclusions and anatomical regions. Due to the limited miRNA expression data obtained from some diseases, analyses to identify consistently differentially expressed miRNA as per anatomy were performed solely with data from MSA, PD and AD.

A total of 32 unique miRNAs were consistently downregulated, while 18 were consistently upregulated across neurodegeneration for the collective analyses performed (Table 2). Across all NDDs, as well as within individual diseases (AD, PD, MSA), the miRNAs displayed differential expression as per the tissue/biofluid from which they were identified post RRA. No miRNAs were commonly DE in the different anatomical regions analyzed across diseases (Figure 5A,B). In AD, PD and MSA, a total of 38, 55 and 58 miRNAs were consistently downregulated and 12, 63 and 34 were upregulated, respectively, for all analyses (Table 2).

Thereafter, the RRA method was used for miRNAs grouped together on the basis of their underlying proteinopathy. The significant miRNA were overlapped post analyses (Figure 5C,D). The tauopathy expression profiles produced a single significant upregulated miRNA post RRA analyses (hsa-miR-100), which was also expressed in both the synuclein and amyloid pathologies. Among the consistently downregulated miRNAs, 39, 16, 1 and 1 miRNAs were unique to α-synuclein, amyloid, TDP and tau pathologies, respectively (Table 3), corresponding to a probable unique expression signature in each proteinopathy. Hsa-miR-132-5p, hsa-miR-375 and hsa-miR-132-3p were the sole miRNAs commonly downregulated across all proteinopathies analyzed. Our analyses revealed 17 and 44 unique miRNAs post RRA specifically upregulated in amyloidopathies and synucleinopathies alone (Table 3).

### 3.4. Gene Set and Pathway Enrichment Analysis

The ranked meta-analytic miRNA datasets obtained from AD, PD and MSA, the collective NDDs as well as individual proteinopathies were subjected to enrichment analysis to identify the pathways that were enriched. Each miRNA-gene association within the enriched miRNA lists was treated as an individual entity. Our results revealed no significant enrichment for any analysis performed for the set criterion of a permutation *p*-value < 0.05.

## 4. Discussion

With the objectives of differentiating between diseases having the same protein pathology, discriminating amongst protein pathologies and investigating if a common mechanism of pathogenesis exists in diverse NDDs, we performed a meta-analysis using the miRNA expression profiles of seven NDDs. Our analyses revealed miRNAs specific to the anatomy, disease and type of pathological protein. Consistent with the literature that implicates their roles in modulating essential neuronal functions, hsa-miR-146a-5p, hsa-miR-100, hsa-miR-132 and hsa-miR-375 were consistently differentially expressed across neurodegeneration and within selective NDDs (Table 2 and Table 3; [85,86,87,88]). Although miRNAs were consistently dysregulated within individual proteinopathies, as well as across neurodegeneration in general, our analyses yielded no functional enrichment unique to a disease, pathology or neurodegenerative mechanism.

Our initial assessment evaluated the clustering of DE miRNAs from several NDDs. These DE miRNAs were obtained from different subcortical and noncortical regions, as well as from biofluids (i.e., different Brodmann areas of the cerebral cortex, the cingulate gyrus, hippocampus, cerebellar cortex, striatum, CSF and serum). Only 3% of miRNAs were commonly downregulated, and even fewer were upregulated between these sources. These results indicate that miRNA dysregulation is primarily tissue- and cell-specific in these diseases. Altogether, the number of up- and downregulated miRNAs unique to each disease was greater than the overlap between them, which is possibly indicative of distinct pathophysiological mechanisms within each disease. However, our clustering analyses suggest that the diverse miRNAs obtained from individual diseases correspond to heterogeneity in the profiling platforms, statistical analysis and lab techniques that were followed across the studies.

A number of miRNAs were consistently differentially expressed and were deemed significant within diseases, proteinopathies and across neurodegeneration with respect to the direction of dysregulation and the tissue/biofluid from which they were identified post RRA analyses. These miRNAs included hsa-miR-206, hsa-miR-24 and hsa-miR-127 (listed in Table 2 and Table 3). However, we should be cautious when drawing inferences from the data. For example, hsa-miR-206 and hsa-miR-24 were identified as significantly differentially expressed across neurodegeneration because they were specifically differentially expressed in PD and MSA, respectively. On the other hand, hsa-miR-127 was consistently differentially expressed in AD and across neurodegeneration. This miRNA was also dysregulated in PD; thus, even though the number of profiles from AD may bias our analyses, hsa-miR-127′s dysregulation in neurodegeneration cannot be attributed to AD alone. Additionally, although different diseases may share a commonly differentially expressed miRNA, its direction of dysregulation can differ in the individual diseases. For instance, hsa-miR-146a-5p was upregulated in PD; however, both up- and downregulation was observed in AD and across neurodegeneration.

Hsa-miR-132-5p, hsa-miR-375 and hsa-miR-132-3p were the only consistently differentially downregulated miRNAs across all pathologies; hsa-miR-100 was the sole miRNA commonly upregulated in amyloidopathies, synucleinopathies and tauopathies. Collectively, these miRNAs impact vital cellular functions. MiR-375 suppresses apoptosis and facilitates neurogenesis in spinal motor neurons [89]. Similarly, miR-100 levels modulate apoptosis in pathologic conditions overexpressing amyloid-β and can regulate inflammatory factors involved in microglia activation [86,90], and miR-132 has been implicated in the extensive regulation of neurons, including but not limited to neurogenesis, neuronal differentiation, neural migration, synaptic plasticity and protection against amyloid-β and tau aggregation and glutamate excitotoxicity [85,86,87,88]. The downregulation of hsa-miR-433 has been implicated in increasing the susceptibility to acquiring PD by failing to repress excessive Fibroblast Growth Factor 20 (FGF20) mRNA translation through complementary binding, which is a protein that has been shown to predispose individuals to PD [91]. Though this miRNA is not yet implicated in other NDDs in the literature, our analyses deemed miR-433 significantly downregulated across neurodegeneration, in AD and in MSA. Thus, these miRNAs’ dysregulation can contribute to neurodegenerative processes in all the pathologies. Conversely, the miRNAs unique to a specific pathology may play a role in determining the location and cell type affected in each disease exhibiting the underlying proteinopathy.

Post RRA, the Venn diagrams revealed minimal clustering between biofluids and other tissue: hsa-miR-127-3p was downregulated and hsa-miR-146a-5p was upregulated in both the noncortical and biofluid data; and hsa-miR-100 was upregulated in both the cortical and biofluid data. As both the downregulation of miR-127-3p and the upregulation of miR-100 downregulate autophagy, their dysregulation may be protective [86,92]. MiR-146a is implicated in many neurodevelopmental disorders and is suspected to contribute to the acquisition of neuronal identities [93]. These miRNAs may be representative of miRNAs with diagnostic potential if they are confirmed to be significantly dysregulated in the specific disease or proteinopathy in both tissue and biofluid.

On assessing the consistently differentially expressed miRNAs post RRA for functional enrichment, no enrichment was found within any disease, proteinopathy and across neurodegeneration. Although the reason for the lack of enrichment within a disease is not clear, we speculate that the molecular functions and biological activities that are modulated may be dependent on cell-specific functions and the cellular location and, hence, may have distinct targets. No enriched pathways in individual proteinopathies and neurodegeneration could be attributed to the activation of distinct pathways and the dysfunction of different regulatory molecules in the individual diseases included.

This study, as a whole, had a number of limitations. Primarily, there exists a clear imbalance in the number of profiles analyzed in certain diseases (Figure 1). Since PSP, FTLD, CBD and LBD have only one or two profiles included for each disease, the results of the analyses conducted are biased towards the diseases with greater numbers of profiles analyzed. Secondly, the differences in the stages of pathology of the analyzed tissue within individual diseases may have resulted in few shared miRNAs due to deficient pathology in some of the tissues analyzed. Moreover, although a disease can manifest two or more pathological proteins, DE miRNAs from analyzed tissue may correspond to only one protein on account of each protein having a different onset, progressing at a different duration and following its own route of progression [11,15]. For example, in AD, the middle temporal gyri are affected in Thal phases I–II and Braak stages IV–V; DE miRNAs from the temporal gyri at a lower Braak stage may only reflect amyloid plaques. Another limitation in identifying the shared pathomechanisms between different proteinopathies via commonly expressed miRNAs arises due to common datasets from shared diseases between pathologies (i.e., a disease manifesting multiple underlying pathological proteins). For example, amyloid plaques and tau tangles are characteristic features of AD; TDP-43 aggregates are also identified in AD patients [10]. Hence, the overlap observed between tauopathies, TDP-43 proteinopathies and amyloidopathies may be primarily attributed to AD. Additionally, we encountered a lack of specificity in the miRNAs detected via profiling and in the research undertaken for each miRNA. The majority of miRNAs have functional 5p and 3p strands. Although 5p and 3p strands display at least partial complementarity, their target sequences across the genome vary and, hence, each individual strand’s targeted binding may regulate different genes and should be studied and accounted for separately.

Altogether this implies that, in addition to expanding the existing research within each disease, an approach to standardize the postmortem intervals for the tissue usage, tissue and cell type, postmortem pathological protein confirmation, extraction methods, lab techniques and profiling platforms is necessary to minimize the variability between studies. An alternative approach to identify the presence of a miRNA expression signature specific to a proteinopathy could be by utilizing biological specimens with validated pathological protein aggregates, thereby preventing the use of a disease’s expression profile for multiple proteinopathies and obtaining an unbiased neurodegenerative signature should their dysregulated miRNAs overlap. A long-term goal would be to validate the pathological stage and make inferences not only on the basis of expression data, but also the tissue and cell type used. These research endeavors would greatly aid and benefit researchers in drawing well-founded conclusions concerning the miRNA expression patterns in NDDs.

## Figures and Tables

**Figure 1 genes-13-01034-f001:**
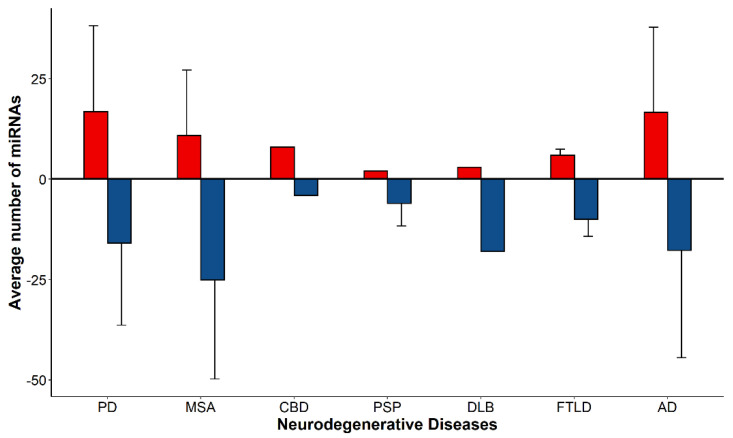
The average number of differentially expressed miRNAs per profile in Parkinson’s disease, multiple system atrophy, corticobasal degeneration, progressive supranuclear palsy, dementia with Lewy bodies, frontotemporal lobar degeneration and Alzheimer’s disease. A total of 17 ± 21, 11 ± 16, 8, 2, 3, 6 ± 1 and 17 ± 21 miRNAs were upregulated in PD, MSA, CBD, PSP, DLB FTLD and AD, respectively. A total of 16 ± 20, 25 ± 25, 4, 6 ± 6, 18, 10 ± 4 and 18 ± 27 miRNAs were downregulated in PD, MSA, CBD, PSP, DLB FTLD and AD, respectively. Data are represented as mean ± SD. N = 9, 8, 1, 2, 1, 2, 22 upregulated miRNA profiles, and 12, 7, 1, 2, 1, 2, 22 downregulated miRNA profiles in PD, MSA, CBD, PSP, DLB FTLD and AD, respectively. The SD observed is lowest in NDDs with fewer profiles considered.

**Figure 2 genes-13-01034-f002:**
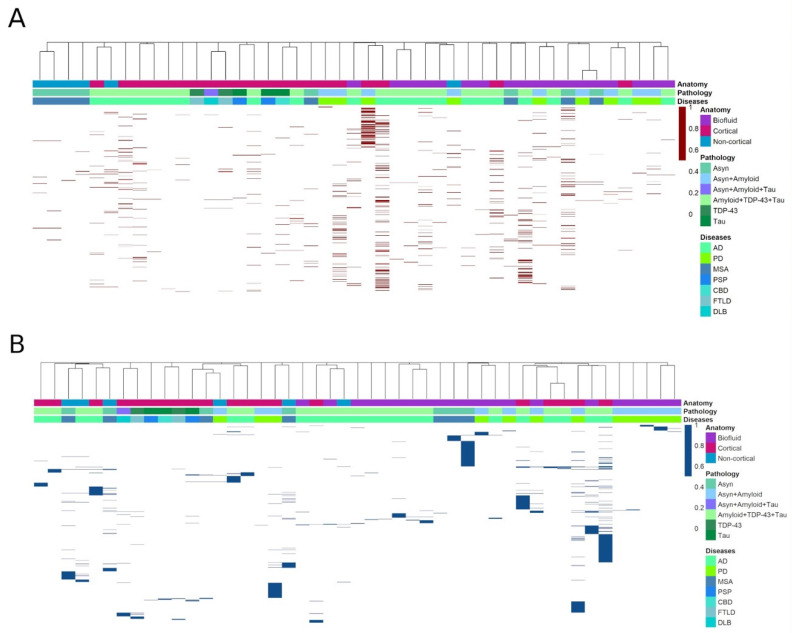
Distribution of upregulated (red) (**A**) and downregulated (blue) (**B**) miRNAs. N = 45 (upregulated) and 47 (downregulated) expression profiles. The X-axis in each heatmap corresponds to a miRNA profile, while the *Y*-axis corresponds to individual miRNAs differentially expressed. The heatmap classifies profiles on the basis of miRNA differential expression in a specific disease, pathology or anatomy. The heatmap demonstrated minimal clustering as per these specified criteria. A greater number of upregulated miRNAs were repeatedly expressed. The distance between profiles was calculated using the asymmetric binary similarity measure.

**Figure 3 genes-13-01034-f003:**
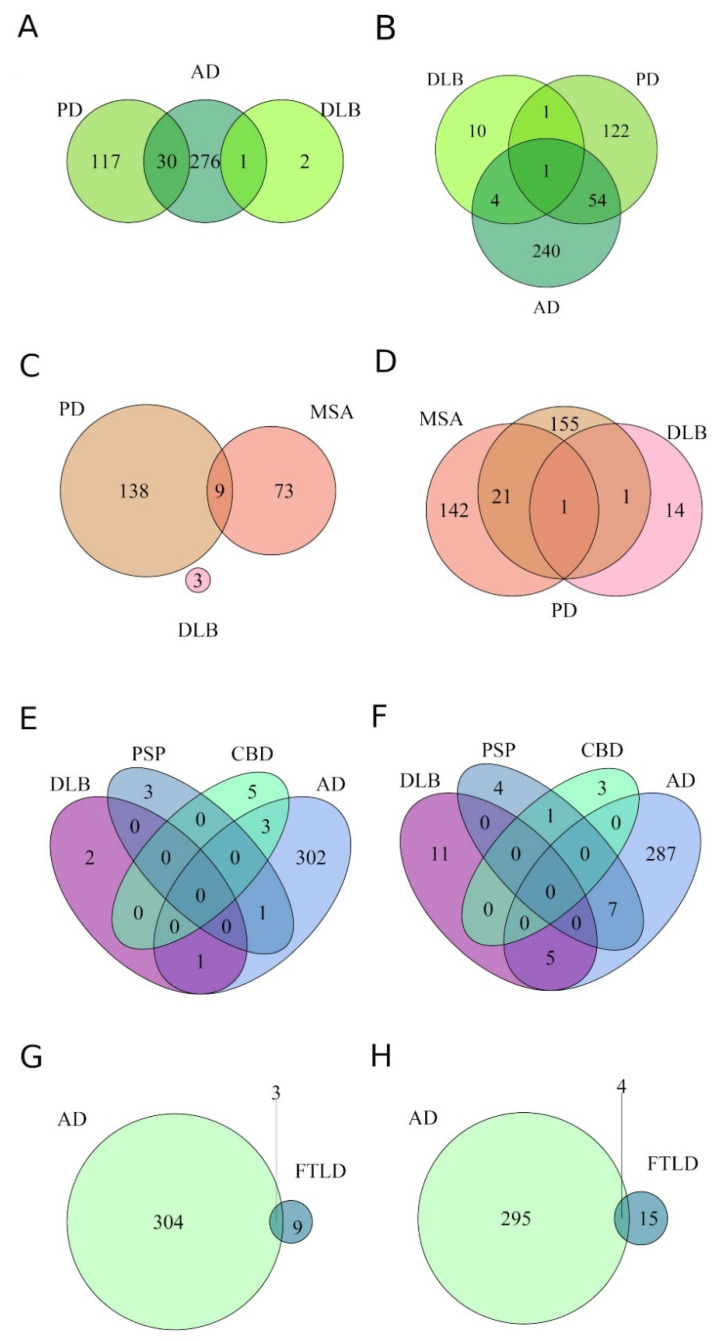
Venn diagrams representing the overlap of miRNAs differentially expressed in individual neurodegenerative diseases, sharing a common underlying protein pathology. Overlap is displayed between amyloidopathies (**A**,**B**), ɑ-synucleinopathies (**C**,**D**), tauopathies (**E**,**F**) and TDP-43 proteinopathies (**G**,**H**). (**A**,**C**,**E**,**G**) demonstrate upregulated miRNAs, while (**B**,**D**,**F**,**H**) demonstrate downregulated miRNAs in the individual proteinopathies. A 0–1% overlap was observed between diseases included within a single proteinopathy.

**Figure 4 genes-13-01034-f004:**
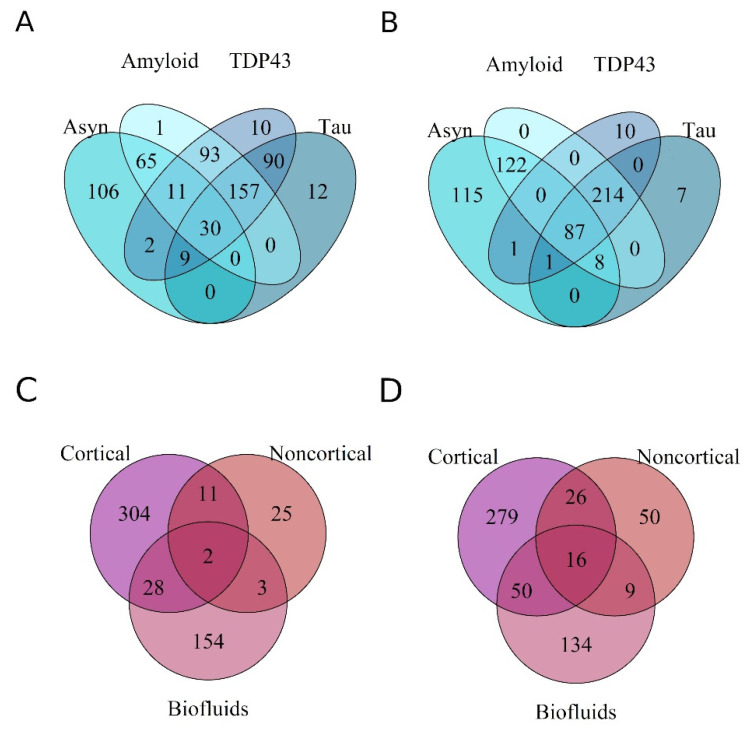
Venn diagrams representing the overlap of miRNAs differentially expressed within a proteinopathy (**A**,**B**), and in a specific anatomical region or biofluid (**C**,**D**). (**A**,**C**) demonstrate upregulated miRNAs, while (**B**,**D**) demonstrate downregulated miRNAs. Few miRNAs were unique to each proteinopathy and anatomy. 15% and 5% of DE miRNAs were commonly down- and upregulated, respectively, between all protein pathologies. Less than 5% of DE miRNAs obtained from the expression profiles were commonly DE in both brain material and biofluids.

**Figure 5 genes-13-01034-f005:**
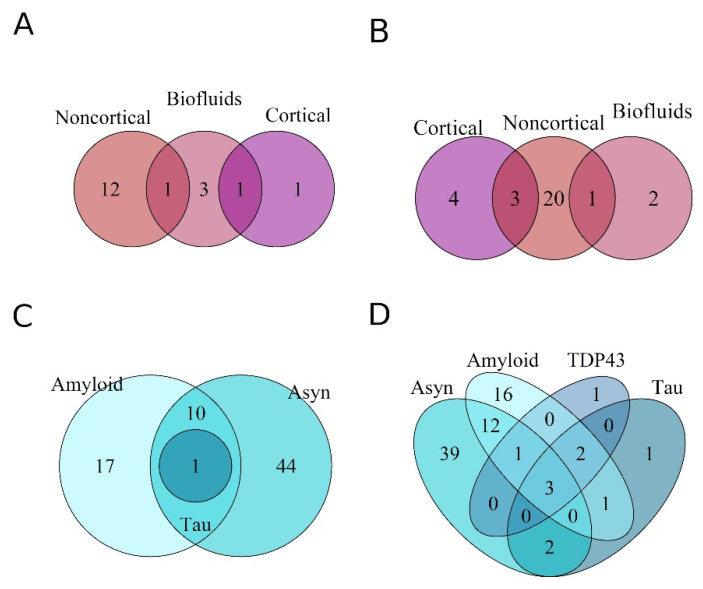
Venn diagrams representing miRNAs consistently differentially expressed post RRA analyses. The overlap between miRNAs from brain tissue and biofluids: upregulated (**A**), downregulated (**B**). Overlap of DE miRNAs between proteinopathies: upregulated (**C**), downregulated (**D**).

**Table 2 genes-13-01034-t002:** List of consistently differentially expressed miRNAs post RRA in AD, PD, MSA and across neurodegeneration.

Upregulated				Downregulated			
Neurodegeneration	AD	PD	MSA	Neurodegeneration	AD	PD	MSA
hsa-miR-146a-5p	hsa-miR-223-5p	hsa-miR-146a-5p	hsa-miR-1290	hsa-miR-100	hsa-miR-132-3p	hsa-miR-132-5p	hsa-miR-129-5p
hsa-miR-24	hsa-miR-548-5p	chr6_novelMiR_46	hsa-miR-4428	hsa-miR-146a-5p	hsa-miR-132-5p	hsa-miR-501-3p	hsa-miR-4440
hsa-miR-206	hsa-miR-1184	hsa-miR-19a-3p	hsa-miR-4736	hsa-miR-127-3p	hsa-miR-375	chr9_novelMiR_225	hsa-miR-128-3p
hsa-miR-223-5p	hsa-miR-34b-3p	hsa-miR-338-3p	hsa-miR-4258	hsa-miR-139-5p	hsa-miR-139-5p	hsa-miR-1294	hsa-miR-4726-5p
hsa-miR-100	hsa-miR-455-3p	hsa-miR-7157-5p	hsa-miR-4708-3p	hsa-miR-136-3p	hsa-miR-210	hsa-miR-16-2-3p	hsa-miR-20b
hsa-miR-1249	hsa-miR-146a-5p	hsa-miR-24-3p	hsa-miR-3622a-5p	hsa-miR-132-3p	hsa-miR-129-5p	hsa-miR-221	hsa-miR-371b-3p
hsa-miR-1290	hsa-miR-501–3p	hsa-miR-9903	hsa-miR-371b-5p	hsa-miR-132-5p	hsa-miR-212-3p	hsa-miR-485-5p	hsa-miR-4726-3p
hsa-miR-4428	hsa-miR-941	hsa-miR-206	hsa-miR-663a	hsa-miR-129-5p	hsa-miR-140-3p	hsa-miR-451b	hsa-miR-433
hsa-miR-24-3p	hsa-miR-1180-3p	hsa-miR-19b-3p	hsa-miR-148b	hsa-miR-20b	hsa-miR-185-5p	hsa-miR-149-5p	hsa-miR-4667-5p
hsa-miR-23a-3p	hsa-miR-152	hsa-miR-30e-3p	hsa-let-7i	hsa-miR-375	hsa-miR-501-3p	hsa-miR-129-5p	hsa-miR-339-5p
hsa-miR-21-5p	hsa-miR-99b	hsa-miR-223	hsa-miR-324-3p	hsa-miR-212-5p	hsa-miR-127-3p	hsa-miR-4772-5p	hsa-miR-4325
hsa-miR-4732-5p	hsa-miR-153	hsa-miR-10a-5p	hsa-miR-522	hsa-miR-138-5p	hsa-miR-136-3p	hsa-miR-150-5p	hsa-miR-24-3p
hsa-miR-199a-5p		hsa-miR-30a-3p	hsa-miR-484	hsa-miR-433	hsa-miR-124-3p	hsa-miR-3159	hsa-miR-370-3p
hsa-miR-151b		hsa-miR-195	hsa-miR-223	hsa-miR-212-3p	hsa-miR-182-5p	hsa-miR-10b-5p	hsa-miR-4728-5p
hsa-miR-219a-2-3p		hsa-miR-24	hsa-miR-4693-3p	hsa-miR-26b	hsa-miR-125b-3p	hsa-miR-1249-3p	hsa-miR-380
hsa-miR-192-5p		hsa-miR-191	hsa-miR-103a	hsa-miR-129-2-3p	hsa-miR-212-5p	hsa-miR-3908	hsa-miR-2392
hsa-miR-30b-5p		hsa-miR-132-5p	hsa-miR-4298	hsa-miR-409-5p	hsa-miR-378a-3p	hsa-miR-3607-3p	hsa-miR-4489
hsa-miR-146b-5p		hsa-miR-1249	hsa-let-7c	hsa-miR-184	hsa-miR-132	hsa-miR-6776-3p	hsa-miR-3912
		hsa-miR-6512-3p	hsa-miR-21-5p	hsa-miR-501-3p	hsa-miR-885-3p	hsa-miR-885-5p	hsa-miR-4722-5p
		hsa-miR-151b	hsa-miR-4732-5p	hsa-miR-484	bta-miR-2487	hsa-miR-3653-5p	hsa-miR-3150a-3p
		hsa-miR-192-5p	hsa-miR-199a-5p	hsa-miR-34c-3p	hsa-miR-184	hsa-miR-1271-5p	hsa-miR-658
		hsa-miR-3153	hsa-miR-219a-2-3p	hsa-miR-149-5p	hsa-miR-34c-3p	hsa-miR-497-5p	hsa-miR-4661-3p
		hsa-miR-1301-3p	hsa-miR-30b-5p	hsa-miR-371b-3p	hsa-miR-128	hsa-miR-3607-5p	hsa-miR-1587
		hsa-miR-146b-5p	hsa-miR-486-5p	hsa-miR-4726-3p	hsa-miR-487b	hsa-miR-484	hsa-miR-4270
		hsa-miR-5193	hsa-miR-3619-3p	hsa-miR-1271-5p	hsa-miR-370	hsa-miR-132-3p	hsa-miR-147b
		hsa-miR-30a-5p	hsa-miR-202	hsa-miR-128-3p	hsa-miR-433	hsa-miR-5009-5p	hsa-miR-3162-5p
		hsa-miR-1322	hsa-miR-24-3p	hsa-miR-150-5p	hsa-miR-138-5p	hsa-let-7d-3p	hsa-miR-4428
		hsa-miR-6850-3p	hsa-miR-612	hsa-miR-1233-3p	hsa-miR-129-2-3p	hsa-miR-129-1-3p	hsa-miR-4800-5p
		hsa-miR-33a-5p	hsa-miR-23a-3p	hsa-miR-1228-3p	hsa-miR-136-5p	hsa-miR-138-5p	hsa-miR-4795-3p
		hsa-miR-191-5p	hsa-miR-4725-3p	hsa-miR-3928-3p	hsa-miR-495-3p	hsa-miR-6826-5p	hsa-miR-4776-5p
		hsa-miR-302b-3p	hsa-miR-199a-3p	hsa-miR-4739	hsa-miR-329	hsa-miR-103a-2-5p	hsa-miR-149-3p
		hsa-let-7d-3p	hsa-miR-181a-5p	hsa-miR-1249-3p	hsa-miR-409-5p	hsa-miR-361-3p	hsa-miR-3939
		hsa-miR-126-5p	hsa-miR-143		hsa-miR-487a	hsa-miR-3123	hsa-miR-671-5p
		hsa-miR-32-5p	hsa-miR-181a-3p		hsa-miR-410	hsa-miR-409-5p	hsa-miR-920
		hsa-miR-373-3p			hsa-miR-543	hsa-miR-1224-5p	hsa-miR-3187-5p
		hsa-miR-1307-5p			hsa-miR-769-5p	hsa-miR-521	hsa-miR-3155
		hsa-miR-1185-1-3p			hsa-miR-219-2-3p	hsa-miR-212-5p	hsa-miR-3622b-5p
		hsa-miR-5701			hsa-miR-425-5p	hsa-miR-129-2-3p	hsa-miR-1228-3p
		hsa-miR-4472				hsa-let-7b-3p	hsa-miR-3928-3p
		hsa-let-7a-3p				hsa-miR-4266	hsa-miR-98
		hsa-miR-1908-3p				hsa-miR-1275	hsa-miR-1915-3p
		hsa-miR-100-3p				hsa-miR-640	hsa-miR-346
		hsa-miR-4673				hsa-miR-1262	hsa-miR-485-5p
		hsa-miR-4524b-3p				hsa-miR-6812-5p	hsa-miR-1233-3p
		hsa-miR-1185-5p				hsa-miR-4506	hsa-miR-4739
		hsa-miR-4268				hsa-miR-1180-5p	hsa-miR-26b
		hsa-miR-4691-3p				hsa-miR-5196-3p	hsa-miR-134-5p
		hsa-miR-1224-3p				hsa-miR-124-5p	hsa-miR-129-2-3p
		hsa-miR-23c				hsa-miR-6768-5p	hsa-miR-1203
		hsa-miR-125b-5p				hsa-miR-4289	hsa-miR-3663-5p
		hsa-miR-6769a-3p				hsa-miR-3942-3p	hsa-miR-1826
		hsa-miR-302d-3p				hsa-let-7i-3p	hsa-miR-584-5p
		hsa-miR-767-5p				hsa-miR-128-2-5p	hsa-miR-19b
		hsa-miR-1273h-3p				hsa-miR-10a-5p	hsa-miR-1910-5p
		hsa-miR-4262				hsa-miR-93-3p	hsa-miR-129-3p
		hsa-miR-4788					hsa-miR-132-3p
		hsa-miR-1285-3p					hsa-miR-675-5p
		hsa-miR-410-3p					hsa-miR-1308
		hsa-miR-1303					
		hsa-miR-4446-3p					
		hsa-miR-6509-3p					
		hsa-let-7e-5p					
		hsa-miR-6736-3p					

**Table 3 genes-13-01034-t003:** List of consistently differentially expressed miRNAs post RRA in Synucleinopathies, Amyloidopathies, Tauopathies and TDP-43 proteinopathies.

Downregulated				Upregulated		
Synucleinopathy	Amyloidopathy	Tauopathy	TDP-43 Proteinopathy	Synucleinopathy	Amyloidopathy	Tauopathy
hsa-miR-16-2-3p	hsa-miR-136-3p	hsa-miR-210	hsa-miR-378a-3p	hsa-miR-24	hsa-miR-455-5p	hsa-miR-100
hsa-miR-129-1-3p	hsa-miR-184			hsa-miR-1290	hsa-miR-32-5p	
hsa-miR-128-3p	hsa-miR-128			hsa-miR-223	hsa-miR-1307-5p	
chr9_novelMiR_225	hsa-miR-34c-3p			hsa-miR-19a-3p	hsa-miR-126-5p	
hsa-miR-484	hsa-miR-487b			hsa-miR-338-3p	hsa-miR-363-3p	
hsa-miR-451b	hsa-miR-370			hsa-miR-4736	hsa-miR-27a-3p	
hsa-miR-98	hsa-miR-136-5p			hsa-miR-206	hsa-let-7f-5p	
hsa-miR-28-5p	hsa-miR-124-3p			hsa-miR-19b-3p	hsa-miR-223-3p	
hsa-miR-10b-5p	hsa-miR-495-3p			hsa-miR-30e-3p	hsa-miR-195-5p	
hsa-miR-26b	hsa-miR-329			hsa-miR-4524b-3p	hsa-miR-142-3p	
hsa-miR-1826	hsa-miR-487a			hsa-miR-1	hsa-miR-150-5p	
hsa-miR-4772-5p	hsa-miR-410			hsa-miR-3907	hsa-let-7i-5p	
hsa-miR-19b	hsa-miR-543			hsa-miR-4262	hsa-miR-362-3p	
mmu-miR-212-5p	hsa-miR-769-5p			hsa-miR-101-3p	hsa-miR-92b-3p	
hsa-miR-1294	hsa-miR-219-2-3p			hsa-miR-1306-3p	hsa-miR-199a-3p	
hsa-miR-1308	hsa-miR-425-5p			hsa-miR-4258	hsa-miR-199b-3p	
hsa-miR-3159				hsa-miR-320b	hsa-miR-200a-3p	
hsa-miR-103a-2-5p				hsa-miR-486-5p		
hsa-miR-371b-3p				chr6_novelMiR_46		
hsa-miR-4726-3p				hsa-miR-7157-5p		
hsa-miR-1228-3p				hsa-miR-34c-3p		
hsa-miR-3928-3p				hsa-miR-1301-3p		
hsa-miR-1915-3p				hsa-miR-612		
hsa-miR-346				hsa-miR-431		
hsa-miR-485-5p				hsa-miR-1322		
hsa-miR-1233-3p				hsa-miR-9903		
hsa-miR-4739				hsa-miR-6512-3p		
hsa-miR-134-5p				hsa-miR-663b		
hsa-miR-1203				hsa-let-7d-3p		
hsa-miR-497-5p				hsa-miR-1272		
hsa-miR-3663-5p				hsa-miR-1185-1-3p		
hsa-miR-584-5p				hsa-miR-296-5p		
hsa-miR-4440				hsa-miR-3153		
hsa-let-7d-3p				hsa-miR-643		
hsa-miR-1910-5p				hsa-let-7a-3p		
hsa-miR-129-3p				hsa-miR-4428		
hsa-miR-361-3p				hsa-miR-21-5p		
hsa-miR-675-5p				hsa-miR-4732-5p		
hsa-miR-125a-5p				hsa-miR-199a-5p		
				hsa-miR-219a-2-3p		
				hsa-miR-30b-5p		
				hsa-miR-3619-3p		
				hsa-miR-202		
				hsa-miR-4725-3p		

## Data Availability

Data will be made available on publication. The data can be requested by emailing the corresponding author. Data will be shared with bona fide researchers after approval of proposals with signed data access agreements as required by, and subject to, institutional and national regulations.

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
