# Peer review of "Differentially Expressed miRNAs in Age-Related Neurodegenerative Diseases: A Meta-Analysis"

_genes, 2022, doi:10.3390/genes13061034_

Round 1
Reviewer 1 Report
Dear authors,
This is an interesting meta-analysis analyzing possible overlapping among miRNA in neurodegenerative diseases. I have some questions:
- Why did you decide to mix arrays with RNA sequencing? Arrays have more background noise and I am wondering if this could affect your results. Won't be better to focus only on RNA sequencing, i.e.?
- There are certain places where it is not clear with data you are using. I.e. "Differential expression of miRNA within proteinopathies" have you distinguished among tissues/fluid samples or have you taken all data? You should clarified that moreover, in the discussion you well pointed out that the differential expression of miRNA is tissue/fluid specific.
- Discussion, more literature review in connection with your results is needed.
Thank you very much.
Reviewer 2 Report
I thoroughly enjoyed reading the article about the meta-analyses of miRNAs in NDD. However, I have comments about improving this article -
1. Line 88 - The authors should write the complete forms of MSA, DLB, CBD, and FTLD before using the abbreviations.
2. Line 211 - Can the authors include a table to talk about the characters of these miRNAs?
3. Line 233 - Can the authors infer anything about the disease from these values of overlapping miRNAs?
4. Line 239 - hsa-miR-433 - What is the biological significance of this miRNA?
5. Line 226 - 270 - This sentence has too much information and sounds difficult for a good comparison. Can the authors break this down into 3 or 4 sentences instead?
6. Line 271 - Is there deeper molecular reasoning behind a higher number of upregulated miRNA than a lower number of downregulated miRNAs?
7. Line 284 - 285 - Are there downregulated miRNAs across all NDDs found in this group? Is there any significance to it?
8. Line 33 - Given the p-value for the Gene set enrichment, what is the ultimate significance of this study
Round 2
Reviewer 1 Report
Dear Authors,
The manuscript improved a lot.
Thank you so much.